# OpenReview forum: "FoeGlass: When Simple In-Context Learning Is Enough for Red Teaming Audio Deepfake Detectors"
_ICLR.cc/2026/Conference — Submitted to ICLR 2026_

### Official Review · Reviewer_rpfF · 2025-10-26

**Soundness:** 2
**Presentation:** 3
**Contribution:** 3
**Rating:** 4
**Confidence:** 4

**Summary:**

This paper proposes a method that controls large language model with chain-of-thought prompting on generating new audio Deepfake attacks to hack the Deepfake detectors, making text-to-speech systems generating difficult outputs, compensating the heavy labor inputs. The real attacker becomes then the LLM itself. A text-based feedback module prior to the detector is set , in order to provide the feedback for the context designer module. The full prompt attempt has been given in the Appendix as examples.

**Strengths:**

1. This paper is very complete and clear in terms of presentation, objective an motivation. It has clear research objective, which is also quite concerning in modern audio Deepfake detection.
2. In terms of originality, this study leverages the LLMs to control the TTS generator, with reward-like feedback scheme. It provides good level of insights to generate novel spoofing attacks, as well as training the defending models.
3. Also thanks to the high transparency and simplicity of the related methods and analysis, this paper potentially has high level of re-producibility, despite the acquisition of LLMs, which can be a varying factor for generating random outputs via inference.

**Weaknesses:**

1. In the related work of research, the author shall cite more about applying audio LLMs for security concern and more specifically, Deepfake detection.
2. The experimental comparison between ASVspoof 5 and the proposed method might be unfair, due to the natural difference between deterministic and non-deterministic approaches. ASVspoof 5 data collection leverages a wide range of collaborators, but each team only contribute for a fixed number of samples within given context. The authors shall compare its work with more in-the-wild fake datasets such as In-the-Wild [1] and Deepfake-Eval-2024 [2], which contains more variations and fake samples. Of course, since we only concerns FNR here, the authors are advised to fetch only fake samples from those datasets, and see whether we can have consistency outperforming results.
3. The absence of real samples. The current modern audio Deepfake detectors mainly has been trained on both real and fake samples, and acquiring solely fake samples may be biased when comparing across different models. This is not that relevant to the main novelty of the work, but including some real samples (e.g. from VCTK) and construct the evaluation set for computing equal error rates can help - the TTS models themselves sometimes will cast bias, and EERs can sidenote this or help finding such issue.
4. The authors are not clear about the prompts themselves in this study - what part of the related prompts are annotated and written by the authors? What are automatically generated from the LLMs, with the instruction from the authors? The authors need to be crustal clear on this, to make the work extendable and more insightful.

[1] Chandra, Nuria Alina et al. “Deepfake-Eval-2024: A Multi-Modal In-the-Wild Benchmark of Deepfakes Circulated in 2024.” ArXiv abs/2503.02857 (2025).
[2] Müller, N., Czempin, P., Diekmann, F., Froghyar, A., Böttinger, K. (2022) Does Audio Deepfake Detection Generalize? Proc. Interspeech 2022, 2783-2787, doi: 10.21437/Interspeech.2022-108

**Questions:**

The questions and concerns from the reviewer has been outlined in the weakness section. Please answer or address the concerns in that section.

**Details Of Ethics Concerns:**

This research has ethical concerns which span as below.

1. Using text-to-speech methods for malicious purposes has been severe concern for years. The authors have only made a general promise on condemning the use and proposing related methods. However, for such framework and methods, licensing and disclaimers are what are actually needed, which is not what the authors have addressed throughout the whole paper.
2. In terms of data licensing, if the author is going to release sample data, the licensing can be a issue to discuss and needs confirm.

---

> ### Author Response · Authors · 2025-11-15
> **Response to Reviewer rpfF (Part 1)**
>
> ---
>
> ---
>
> # **Summary of the Review & Response**
>
> ---
>
> ---
>
> We thank the reviewer for their careful and constructive evaluation. We appreciate that they find the paper *complete*, *clear*, and *well-motivated* with a *clear research objective* in modern audio deepfake detection (S1), recognize the *originality in leveraging LLMs to control TTS with reward-like feedback* to generate novel spoofing attacks and strengthen defenses (S2), and highlight the **t***ransparency, simplicity, and reproducibility* of our methods and analyses (S3).
>
> **Primary concerns** and questions are regarding:
>
> **1)** fairness and breadth of comparison with ASVspoof5, and the suggestion to include more in-the-wild fake datasets (W2)
>
> **2)** the absence of real samples in evaluation (W3)
>
> **We address:**
>
> **1)** by clarifying that ASVspoof5 is a **widely used modern benchmark** for audio deepfake detection, and that we compare only on subsets using the same TTS models as in our experiments to isolate the effect of the **sampling strategy** rather than TTS strength. We also explain that in-the-wild datasets like Deepfake-Eval-2024 and other online collections have **unknown TTS generators**, making them unsuitable for the TTS-specific FNR comparison in Table 2.
>
> **2)** by emphasizing that our goal is **not to benchmark or rank ADDs**, but to evaluate FoeGlass’s ability to generate fake samples that **fool** them. Accordingly, we focus on false negatives (attack success rate) rather than full real+fake evaluation and EER, which would be required for a detector comparison study. Our results thus quantify robustness specifically **against FoeGlass attacks**, not overall detector performance.
>
> In the following, we address each of the concerns raised by the reviewer in detail.
>
> ---
>
> ---
>
> # Response to Weaknesses
>
> ---
>
> ---
>
> > **W1. In the related work of research, the author shall cite more about applying audio LLMs for security concern and more specifically, Deepfake detection.**
> >
>
> We thank the reviewer for this suggestion. Our work focuses on red-teaming ADD models rather than proposing a new detector, so given space constraints we centered the related work on adversarial perturbations, natural adversarial examples, and prompt optimization. If there are specific relevant works the reviewer finds missing, we would be happy to consider and cite them in a revised version.
>
> ---
>
> > **W2. The experimental comparison between ASVspoof 5 and the proposed method might be unfair, due to the natural difference between deterministic and non-deterministic approaches. ASVspoof 5 data collection leverages a wide range of collaborators, but each team only contribute for a fixed number of samples within given context. The authors shall compare its work with more in-the-wild fake datasets such as In-the-Wild [1] and Deepfake-Eval-2024 [2], which contains more variations and fake samples. Of course, since we only concerns FNR here, the authors are advised to fetch only fake samples from those datasets, and see whether we can have consistency outperforming results.**
> >
>
> We respectfully ask the reviewer for clarification on what deterministic and non-deterministic approaches mean in this context.  We have interpreted their comment as follows.  The reviewer feels that ASVspoof5 does not have a sufficient variety of fake samples for a fair comparison against any other method of generating fake samples.  As ASVspoof5 is often used as a modern and comprehensive benchmark for audio deepfake detection, we consider it is an appropriate source of data to compare our automated generation method against.  In particular, we restrict our comparisons within ASVspoof5 to the subsets that use exactly the TTS models for our experiments, ensuring a fair comparison in measuring the resulting FNR.
>
> Our results ultimately show that indeed, there is a gap to be filled in ASVspoof5 data coming from particular TTS systems that FoeGlass can address. As our visualization in Figure 4b and quantitative results in Table 2 demonstrate, the FoeGlass samples occupy distinct regions of the feature space missed by the ASVspoof5 data and which might not be discovered without our method. By using these samples to augment any training set, we can harden a detector against its vulnerabilities.
>
> We have considered comparing FoeGlass-generated samples with other datasets such as Deep-Eval-2024 and In-the-Wild datasets but the data in these datasets are gathered from online data and for the fake split, the text-to-speech (TTS) model is unknown while in Table 2, we are presenting the comparison between FNRs with respect to the TTS model to isolate the effect of sampling approach and not the strength of TTS model.

---

> ### Author Response · Authors · 2025-11-15
> **Response to Reviewer rpfF (Part 2)**
>
> > **W3. The absence of real samples. The current modern audio Deepfake detectors mainly has been trained on both real and fake samples, and acquiring solely fake samples may be biased when comparing across different models. This is not that relevant to the main novelty of the work, but including some real samples (e.g. from VCTK) and construct the evaluation set for computing equal error rates can help - the TTS models themselves sometimes will cast bias, and EERs can sidenote this or help finding such issue.**
> >
>
> We agree with the reviewer that if the purpose is to compare ADDs model’s performance, we need to use both real and fake samples. However, we would like to clarify that the scope of our work is not to perform an overall comparison of ADD models. In the paper, we instead evaluate the effectiveness of FoeGlass in generating fake samples that fool and break these models. The comparison provided in the paper is based on ADDs robustness against FoeGlass attacks (false negatives) and not their overall performance.
>
> ---
>
> > **W4. The authors are not clear about the prompts themselves in this study - what part of the related prompts are annotated and written by the authors? What are automatically generated from the LLMs, with the instruction from the authors? The authors need to be crustal clear on this, to make the work extendable and more insightful.**
> >
>
> As mentioned in the paper, the provided prompt to the LLM in each iteration contains: 1) instruction prompt, 2) history, 3) FN samples. The instruction prompt is written by the authors and it is fixed, the history contains last k prompts generated by LLM, and the FN samples are the highest score successful samples generated by the LLM. For more details, please refer to overview Figure in Fig. 2, Section 3, and the prompt examples in Appendix.
>
> ---
>
> ---
>
> ## Response to Ethical Concerns
>
> ---
>
> ---
>
> > **This research has ethical concerns which span as below. 1) Using text-to-speech methods for malicious purposes has been severe concern for years. The authors have only made a general promise on condemning the use and proposing related methods. However, for such framework and methods, licensing and disclaimers are what are actually needed, which is not what the authors have addressed throughout the whole paper. 2) In terms of data licensing, if the author is going to release sample data, the licensing can be a issue to discuss and needs confirm.**
> >
>
> Thank you for raising these ethical points. In the section 7 of the main paper (Ethical Considerations) we strongly condemned the malicious use of FoeGlass and this can serve as a disclaimer. We are committed to ICLR guidelines and will incorporate the necessary licensing that the ethical reviews ask.
>
>
> ---
>
> ---
>
> **Please consider raising the scores if we address your concerns. If you have more questions, we would be happy to answer them.**
>
> ---
>
> ---

---

### Official Review · Reviewer_6vBc · 2025-10-27

**Soundness:** 2
**Presentation:** 3
**Contribution:** 2
**Rating:** 2
**Confidence:** 5

**Summary:**

This paper proposes FoeGlass, a black-box adversarial attack method targeting audio deepfake detection (ADD) systems. FoeGlass explores the input space of text-to-speech (TTS) models through large language model (LLM)-based sampling guided by multiple feedback signals. The effectiveness and transferability of FoeGlass are verified across several TTS–ADD model pairs. This work demonstrates that FoeGlass can identify blind spots in the latent space of ADD systems and cause transcript-level detection failures.

**Strengths:**

- The attacker LLM is well-designed. Its prompt incorporates the history of failed and successful attacks as well as diversity-oriented feedback. The effectiveness of this design is demonstrated in the ablation studies.
- The experiments are comprehensive, showing that FoeGlass successfully discovers blind regions that remain challenging for ADD systems.
- The paper is generally well-written and logically consistent.

**Weaknesses:**

- The paper lacks citations for closely related prior works. Although the authors claim to be the first to design such a transcript-level adversarial method for ADD, I found two highly related papers presenting very similar ideas [1, 2]. The authors should discuss differences and clearly position their contribution relative to prior works. Furthermore, compared to FoeGlass, [1] adopts a greedy optimization approach to generate textual perturbations for important words while preserving semantics. This strategy arguably represents a more realistic and practically feasible attack scenario.
- The evaluation focuses primarily on ADD models that lack pretrained self-supervised learning (SSL) components. These models always perform worse than SSL-based ADD models. I am curious about how FoeGlass would perform on SSL-based ADD models. In addition, as a comparison with [1], evaluation results on commercial detectors are needed.

[1] Nguyen, Binh, et al. "What You Read Isn't What You Hear: Linguistic Sensitivity in Deepfake Speech Detection." arXiv preprint arXiv:2505.17513 (2025).

[2] Turing's Echo: Investigating Linguistic Sensitivity of Deepfake Voice Detection via Gamification.

**Questions:**

- I am concerned about the amount of LLM prompt usage required to generate a successful adversarial audio sample. I believe such an evaluation metric would be important for assessing real-world practicality.
- The visualization of ADD blind spots is persuasive, but this aspect is not further explored. I am interested in what specific factors within FoeGlass-generated samples cause these blind spots. It would be helpful if the authors could suggest strategies for generating audio samples that better cover the overall latent space.

---

> ### Author Response · Authors · 2025-11-15
> **Response to Reviewer 6vBc (Part 1)**
>
> ---
>
> ---
>
> # **Summary of the Review & Response**
>
> ---
>
> ---
>
> We thank the reviewer for their feedback. We are pleased that they find our method to be *well-designed* and *effective* (S1), the experiments to be *comprehensive* (S2), and the manuscript to be *well-written* and *logically consistent* (S2).
>
> **Primary concerns** and questions are regarding:
>
> **1)** some missing concurrent related works (W1)
>
> **2)** the choice of evaluated ADD models (W2)
>
> **We address:**
>
> **1)** by clarifying that the two mentioned papers by the reviewer are **concurrent works** (one even published after the ICLR 2026 deadline) and will be added and discussed in the related work. We explain the fundamental differences in their aim and settings with FoeGlass and highlight that these methods thus target different objectives and threat models.
>
> **2)** by explaining that our ADD choices were guided by **state-of-the-art performance** and have been expanded (following Reviewer uoRF) to include strong open-source models from the Speech-DF-Arena leaderboard, on which FoeGlass remains effective. For **commercial detectors**, we note practical constraints: [1] evaluates anonymized APIs that cannot be reproduced, and most companies are understandably reluctant to permit external red-teaming with public reporting. We therefore advocate for **internal application of FoeGlass** by model owners rather than public benchmarking.
>
> In the following, we address each of the concerns raised by the reviewer in detail.
>
> ---
>
> ---
>
> # Response to Weaknesses
>
> ---
>
> ---
>
> > **W1. The paper lacks citations for closely related prior works. Although the authors claim to be the first to design such a transcript-level adversarial method for ADD, I found two highly related papers presenting very similar ideas [1, 2]. The authors should discuss differences and clearly position their contribution relative to prior works. Furthermore, compared to FoeGlass, [1] adopts a greedy optimization approach to generate textual perturbations for important words while preserving semantics. This strategy arguably represents a more realistic and practically feasible attack scenario.**
> >
>
> Thank you for providing these related works. However, both of the mentioned prior works are considered concurrent work under the ICLR review guidelines as [1] published at EMNLP in November 2025 which is after ICLR 2026 submission deadline, and [2] is published in August 2025. Although they are concurrent works, we will include them in our prior works and provide a discussion on their similarities and differences with FoeGlass.
>
> In summary, the objective and settings of the mentioned papers are completely different from FoeGlass. As mentioned in page 3 (objective function box) of [1], works [1] and similarly [2] focus on **locally perturbing a given transcript** (finding minimal, semantics-preserving edits that flip the detector’s decision), akin to classic ε-ball adversarial examples. By contrast, FoeGlass does **not** start from a fixed transcript; it **explores the broader prompt space** to discover new regions that fool the detector and can be used to **augment training data and strengthen detectors** (Table 3). As highlighted, these methods serve different purposes than FoeGlass.
>
> ---
>
> > **W2. The evaluation focuses primarily on ADD models that lack pretrained self-supervised learning (SSL) components. These models always perform worse than SSL-based ADD models. I am curious about how FoeGlass would perform on SSL-based ADD models. In addition, as a comparison with [1], evaluation results on commercial detectors are needed.**
> >
>
> Thank you for the suggestion. Our choice of ADD models was guided by state-of-the-art performance and has been expanded (following Reviewer uoRF’s comment) to include top entries from the Speech-DF-Arena leaderboard, where FoeGlass remains effective at uncovering failure modes.
>
> Regarding commercial systems, [1] evaluates two proprietary APIs (API-A and API-B) whose identities are not disclosed, making it impossible for us to reproduce those experiments with FoeGlass.
>
> Moreover, we faced a significant barrier for evaluating FoeGlass on commercial ADDs: most companies are understandably reluctant to allow external red teaming of their proprietary models, particularly with public reporting of identified vulnerabilities. Therefore, while we cannot currently provide evaluations against commercial systems, we strongly advocate for the owners of these models to privately apply FoeGlass to identify their models' failure cases and enhance their robustness in a controlled, internal environment.

---

> > ### Author Response · Authors · 2025-11-15
> > **Response to Reviewer 6vBc (Part 2)**
> >
> > ---
> >
> > ---
> >
> > # Response to Questions
> >
> > ---
> >
> > ---
> >
> > > **Q1. I am concerned about the amount of LLM prompt usage required to generate a successful adversarial audio sample. I believe such an evaluation metric would be important for assessing real-world practicality.**
> > >
> >
> > In our experiments, we report **attack success rate**, defined as the fraction of LLM-generated transcripts that yield successful adversarial audio. This metric directly determines the expected prompt usage: with success rate $s$, the expected number of LLM calls per successful attack is $\dfrac{1}{s}$ (e.g., a 98% success rate means roughly **1.02 prompts per successful adversarial sample**), which provides a clear measure of real-world cost.
> >
> > ---
> >
> > > **Q2a. The visualization of ADD blind spots is persuasive, but this aspect is not further explored. I am interested in what specific factors within FoeGlass-generated samples cause these blind spots.**
> > >
> >
> > In addition to visualizing the clustering and coverage of FoeGlass generated data in the WavLM embedding space, in Figure 4a we provide portions of the transcripts that are similar semantically within each of the clusters.  Since FoeGlass explicitly is searching for vulnerabilities resulting from different transcripts, this suggests these language features, combined with the TTS method’s processing of them, are resulting in the ADD model’s discovered blind spots.
> >
> > ---
> >
> > > **Q2b.  It would be helpful if the authors could suggest strategies for generating audio samples that better cover the overall latent space.**
> > >
> >
> > Our objective in designing attacks is to have high success rate while maintaining high diversity and preventing repeating the same successful attack. To exclusively optimize for latent space coverage, there are several studies in representation learning of image modality such as using  Determinantal Point Process (DPP) [R1] to unconditionally diversify the generated samples, these could be adapted to audio generation as well.
> >
> > [R1] Kulesza, Alex, and Ben Taskar. "Determinantal point processes for machine learning." *Foundations and Trends® in Machine Learning* 5.2–3 (2012): 123-286.
> >
> > ---
> >
> > ---
> >
> > # **On the Overall Assessment and Rating**
> >
> > ---
> >
> > ---
> >
> > Given that the reviewer’s main concerns center on **related work that is concurrent with our submission** (which we now explicitly acknowledge and situate) and on additional evaluation axes that we discuss and partially address, and in light of their positive assessment of our method, experiments, and writing, we respectfully believe that a **score of 2 (reject)** is not fully aligned with the substance of the concerns raised. We therefore kindly ask the reviewer to **reconsider their rating** in view of our clarifications, the added discussion of concurrent work, and the strength and breadth of our empirical results.

---

### Official Review · Reviewer_uoRF · 2025-10-31

**Soundness:** 2
**Presentation:** 3
**Contribution:** 2
**Rating:** 4
**Confidence:** 5

**Summary:**

This paper proposes FoeGlass, an automated red-teaming framework for Audio Deepfake Detection systems. FoeGlass uses in-context learning from a black-box reasoning LLM to generate natural adversarial audio samples by only optimizing input transcripts to TTS models.

**Strengths:**

1. FoeGlass introduces an automated red-teaming pipeline that leverages reasoning LLMs and iterative feedback to synthesize natural adversarial audio. This is a useful direction because it avoids hand-crafted perturbations and does not require labeled attack examples.

2. The writing is good and easy to follow.

**Weaknesses:**

1. The citation format/style throughout the paper is inconsistent and should be revised. For example, the very first sentence has an incorrect citation format, and similar issues appear in several other places.
2. The audio deepfake detection models evaluated in the paper are not state-of-the-art. There are stronger and more recent models available on [Hugging Face’s Speech-DF-Arena](https://huggingface.co/spaces/Speech-Arena-2025/Speech-DF-Arena). Given this, I’m not sure how well the proposed method would perform on more advanced models.
3. Also as shown in Table 2, it is interesting to see that, for the AST model, the FNR on ASVspoof5 is even higher than that on the FoeGlass-generated xTTS-v2 data. This further raises concerns about the generalizability of the proposed attack across different ADD models.
4. It’s unclear how the paper defines and calculates the attack success rate. In audio deepfake detection, Equal Error Rate (EER) is a more standard and interpretable metric, as it identifies the point where FNR equals FPR. What threshold is used in the evaluation? If the authors used a fixed threshold of 0.5 to distinguish real and fake audio, please justify why. In practice, different models trained on different datasets typically require threshold tuning, so using a uniform threshold may not yield fair or comparable results. Clarifying this would make the evaluation more convincing.

**Questions:**

1. The paper would benefit from a deeper discussion of why models such as VIT-Melspectrogram and ASV-Spoof5 exhibit better robustness to the proposed attack. Understanding the specific failure cases of FoeGlass could provide more insights valuable insights.
2. It’s not entirely clear how diversity scores and context lengths influence the attack’s performance. Have the authors conducted any ablation studies or sensitivity analyses?
3. In addition, since TTS models such as xTTS-v2 support multiple voices. Did the authors use different voices during red-teaming? Additionally, was there any observable difference in ASR performance across male/female voices, various accents, or languages? I don't see the details of how the attack generates the adversarial audio.
4. While the results may not seem sound to me, the proposed red-teaming framework could be a potential adversarial component to help generate challenging fake audio samples to train the detectors. Have the authors explored if FoeGlass could improve the detection performance?

---

> ### Author Response · Authors · 2025-11-15
> **Response to Reviewer uoRF (Part 1)**
>
> ---
>
> ---
>
> # **Summary of the Review & Response**
>
> ---
>
> ---
>
> We thank the reviewer for their feedback. We are pleased that the reviewer finds our method to be *useful automated red teaming* (S1) and the manuscript to be *well-written* and *easy to follow* (S2).
>
> **Primary concerns** and questions are regarding:
>
> **1)** citation style (W1)
>
> **2)** the choice of evaluated ADD models (W2)
>
> **3)** the definition and evaluation of attack success rate and the choice of threshold (W4)
>
> **4)** whether FoeGlass can also **improve** detection when used as a data generator (Q4)
>
> **We address:**
>
> **1)** by revising the manuscript to enforce a **consistent citation style** throughout.
>
> **2)** by evaluating FoeGlass on new top open-source models, released after the ICLR deadline, and showing substantial gains in attack success (+18.4% and +26.6%) in **Table R1**, while clarifying that our existing ADD set already covers recent and competitive models.
>
> **3)** by formally defining **attack success rate** as the FNR on TTS-generated samples under a fixed decision rule, justifying a fixed threshold of 0.5 as reflecting a realistic black-box, deployed-detector setting. We further support robustness by showing a bimodal score distribution for FoeGlass attacks in **Fig. 8**, indicating insensitivity to threshold changes.
>
> **4)** by pointing out that we already explore the effectiveness of FoeGlass in improving ADDs in **Table 3**, where fine-tuning RawNetLite and AASIST on FoeGlass-generated samples improves robustness by **41%** and **15%**, respectively, showing that FoeGlass both exposes vulnerabilities and yields challenging training data that strengthens ADDs (Q4).
>
> In the following, we address each of the concerns raised by the reviewer in detail.
>
> ---
>
> ---
>
> # Response to Weaknesses
>
> ---
>
> ---
>
> > **W1. The citation format/style throughout the paper is inconsistent and should be revised. For example, the very first sentence has an incorrect citation format, and similar issues appear in several other places.**
> >
>
> Thank you for pointing this out. We have carefully revised the manuscript to use a consistent citation style throughout.
>
> > **W2. The audio deepfake detection models evaluated in the paper are not state-of-the-art. There are stronger and more recent models available on [Hugging Face’s Speech-DF-Arena](https://huggingface.co/spaces/Speech-Arena-2025/Speech-DF-Arena). Given this, I’m not sure how well the proposed method would perform on more advanced models.**
> >
>
> Thank you for pointing us to this leaderboard. As noted there, only a few top-performing open-source models are currently available for integration into our paper, and the strongest among them, **DF_Arena_1B_V_1** and **DF_Arena_500M_V_1**, were released 26 and 11 days ago, respectively (i.e., after the ICLR 2026 submission deadline). While these models are considered concurrent work under the ICLR guidelines, we have nevertheless evaluated FoeGlass against them to assess its effectiveness on these stronger detectors. The results are reported in **Table R1**.
>
> Despite the robustness of DF_Arena_500M and DF_Arena_1B to unconditional Kokoro-generated samples, FoeGlass substantially increases the attack success rate on both models by **18.4%** and **26.6%**, respectively. We incorporated these two rows into the existing Table 5 in Appendix C.
>
> We also emphasize that not all recent audio deepfake detectors (ADDs) are yet included in the leaderboard. For example, **RawNetLite**, which we already evaluate in our paper, is a very recent model (published in April 2025). Our selection of ADDs was designed to cover (i) models trained on state-of-the-art spoofing datasets such as ASVSpoof5, (ii) detectors using a diverse set of visualization features (Constant-Q, MelSpectrogram, MFCC; Table 1), and (iii) raw-audio models such as RawNet2, RawNetLite, and AASIST (Table 5 in Appendix C). This design provides a broad and representative evaluation of FoeGlass against competitive, modern ADDs.
>
> **Table R1:** The average detection accuracy of ADD-TTS systems trained with different backbones.
> Results are shown for attacks generated with unconditional sampling and with FoeGlass. All numbers are in %.
>
> | **ADD-TTS** | **Unconditional Sampling** | **FoeGlass** |
> | --- | --- | --- |
> | **RawNet2-xTTS** | 98.6 | 99.4 |
> | **RawNet2-Kokoro** | 43.4 | 78.4 |
> | **RawNet2-VITS** | 100.0 | 100.0 |
> | **RawNetLite-xTTS** | 100.0 | 100.0 |
> | **RawNetLite-Kokoro** | 88.0 | 90.6 |
> | **RawNetLite-VITS** | 49.6 | 91.8 |
> | **AASIST-xTTS** | 15.2 | 42.2 |
> | **AASIST-Kokoro** | 100.0 | 100.0 |
> | **AASIST-VITS** | 4.5 | 31.6 |
> | **DF_Arena_500M-Kokoro** | 0.0 | 18.4 |
> | **DF_Arena_1B-Kokoro** | 0.6 | 27.2 |

---

> > ### Author Response · Authors · 2025-11-15
> > **Response to Reviewer uoRF (Part 2)**
> >
> > > **W3.  Also as shown in Table 2, it is interesting to see that, for the AST model, the FNR on ASVspoof5 is even higher than that on the FoeGlass-generated xTTS-v2 data. This further raises concerns about the generalizability of the proposed attack across different ADD models.**
> > >
> > Thank you for this observation. We would like to emphasize that in **7 out of 8** ADD models in Table 2, FoeGlass achieves a higher FNR than both ASVspoof5 and unconditional sampling, indicating strong generalizability across architectures. The only exception is AST on the ASVspoof5 xTTS-v2 subset, where ASVspoof5 already yields a good FNR. In this case, FoeGlass still improves over unconditional sampling with the same TTS model, but the ASVspoof5 sampling strategy appears to already cover AST’s main failure modes, leaving limited additional headroom. Overall, the consistent gains in **7/8 cases support that FoeGlass generalizes across different ADD models** rather than overfitting to a particular detector.
> >
> > ---
> >
> > > **W4.  It’s unclear how the paper defines and calculates the attack success rate. In audio deepfake detection, Equal Error Rate (EER) is a more standard and interpretable metric, as it identifies the point where FNR equals FPR. What threshold is used in the evaluation? If the authors used a fixed threshold of 0.5 to distinguish real and fake audio, please justify why. In practice, different models trained on different datasets typically require threshold tuning, so using a uniform threshold may not yield fair or comparable results. Clarifying this would make the evaluation more convincing.**
> > >
> > Thank you for raising this point. In our experiments, **attack success rate** is defined as the fraction of TTS-generated samples that are misclassified as real by the detector, i.e., the false negative rate (FNR) on attack-generated audio under a fixed decision rule. This directly captures the attacker’s objective: to increase type-II errors of a deployed ADD.
> >
> > We use a fixed decision threshold of **0.5** on the detector’s output score. This models a realistic black-box setting in which the attacker cannot tune or sweep the detector’s threshold; instead, the operating point is fixed by the defender prior to deployment. While **EER** is standard for benchmarking **detectors by varying the threshold, our focus is on the vulnerability of a fixed*,* deployed detector, so we measure FNR (attack success rate) at this fixed operating point rather than EER.
> >
> > Importantly, all of our comparisons are **within-detector**: FoeGlass is always evaluated against unconditional sampling using the *same* model and threshold. Thus, our conclusions rely on relative changes in FNR, not on cross-model calibration, and are therefore not sensitive to the particular choice of 0.5.
> >
> > To further assess robustness to the threshold, we analyzed the distribution of detector scores for FoeGlass attacks (Fig. 8 in the revised paper) and observed a strongly bimodal pattern, with scores clustered near 0 (clear failures) and 1 (clear successes). This indicates that successful attacks are far from the decision boundary and would remain successful over a wide range of reasonable thresholds, making the reported attack success rates stable with respect to threshold variations.
> >
> > ---
> > ---
> > # Response to Questions
> > ---
> > ---
> > > **Q1. The paper would benefit from a deeper discussion of why models such as VIT-Melspectrogram and ASV-Spoof5 exhibit better robustness to the proposed attack ...**
> > >
> > Thank you for this suggestion. According to Table 1, for all the TTS-ADD pairs, FoeGlass (whether warm or cold start) outperforms the unconditional sampling baseline. In some specific cases, such as AST-ASVspoof5 targeted by xTTS-v2 and VITS, we see the success rate for all attackers is low, indicating the inherent robustness of these models, partly due to the presence of the TTS models' data in their training/validation sets. We will clarify this in the paper by explicitly discussing these cases.
> >
> > ---
> >
> > > **Q2. It’s not entirely clear how diversity scores and context lengths influence the attack’s performance. Have the authors conducted any ablation studies or sensitivity analyses?**
> > >
> > Our existing results already form a step-wise ablation presented in Table 4 of Sec. B in the appendix:
> >
> > 1. The gain of FoeGlass (No Div. Feedback) over Unconditional Sampling shows the impact of our **context design.**
> > 2. The gain of the full FoeGlass over FoeGlass (No Div. Feedback) demonstrates the benefit of **diversity feedback in the context design**.
> >
> > Furthermore, we investigated the effect of CoT in the provided context. As shown in **Table R2**, CoT is crucial, improving the success rate by up to **27%**.
> >
> > Regarding context length, our experiments show that the success rate is not highly sensitive to context window size; a larger window provides only a slight improvement. We thus chose a context window size of 40 for efficient GPU utilization.

---

> > > ### Author Response · Authors · 2025-11-15
> > > **Response to Reviewer uoRF (Part 3)**
> > >
> > > > **Q3.  In addition, since TTS models such as xTTS-v2 support multiple voices. Did the authors use different voices during red-teaming? Additionally, was there any observable difference in ASR performance across male/female voices, various accents, or languages? I don't see the details of how the attack generates the adversarial audio.**
> > > >
> > >
> > > Thank you for the question. In FoeGlass, we **intentionally keep all TTS generation parameters (including voice) fixed** and only optimize over the **transcript**. The LLM proposes transcripts based on past attempts and ADD feedback; these transcripts are then passed to the TTS model to generate attack audio. This pipeline is described in Section 3, illustrated in Figure 2, and formalized in Algorithm 1.
> > >
> > > This design isolates a key property of the target ADD: **invariance to semantic content**. Ideally, an ADD should be robust to what is being said; our results show that current models are not. We did **not** vary voices, gender, accents, or languages in this work. We agree this is an important axis of robustness and will explicitly note in the conclusion that extending FoeGlass to search over speaker and style parameters is a valuable direction for future work.
> > >
> > > ---
> > >
> > > > **Q4. While the results may not seem sound to me, the proposed red-teaming framework could be a potential adversarial component to help generate challenging fake audio samples to train the detectors. Have the authors explored if FoeGlass could improve the detection performance?**
> > > >
> > >
> > > Thank you for the suggestion. We have already explored this use case in the submitted paper: in **Table 3 (p. 8)**, we fine-tune **RawNetLite** and **AASIST** on FoeGlass-generated samples and observe robustness improvements of **41%** and **15%**, respectively, showing that FoeGlass can indeed be used to generate challenging training data that strengthens detectors.
> > >
> > > We also respectfully disagree with the concern about soundness of results. As shown in **Table 1** (main paper) and **Table 5** (Appendix), FoeGlass consistently achieves substantially higher attack success rates (up to **+94%** over unconditional sampling) across diverse ADD architectures. Together with the fine-tuning gains in Table 3, this indicates that FoeGlass both exposes genuine vulnerabilities and provides a practical tool for improving audio deepfake detection.
> > >
> > > ---
> > >
> > > ---
> > >
> > > **Please consider raising the scores if we address your concerns. If you have more questions, we would be happy to answer them.**
> > >
> > > ---
> > >
> > > ---

---

### Author Response · Authors · 2025-11-20
**Request for Further Reviewer Engagement**

Dear Reviewers,

Thank you for your feedback on our paper.

Our detailed rebuttal was released **5 days ago**, featuring new experimental results and clarifications that directly address the points raised. You may also find our responses to the other reviewers helpful.

We kindly encourage you to share any remaining questions or thoughts, as it would help to make a more informed decision on the paper.

If our rebuttal and revisions have resolved your initial concerns, we would be very grateful if you would consider updating your evaluation.

Thank you again for your time.

---

### Comment · Area_Chair_Ta1J · 2025-11-27

Dear Reviewers,

I noticed that the authors have submitted their rebuttals to your reviews . However, there has been no further engagement or response from your side.

The author-reviewer discussion phase is a critical part of the review process. The authors have put significant effort into addressing your concerns. It is essential that you read their response and acknowledge it.

Please take a moment to:

1. Read the authors' rebuttal carefully.
2. Post a reply indicating whether their response has resolved your concerns.
3. Update your score if appropriate, or explain why your original assessment stands.

Thank you for your immediate attention to this matter.

Best regards,

Area Chair

---

### Meta-Review · Area_Chair_M18G · 2026-01-03

**Summary:**

Reviewers generally found the paper clearly written and the engineering pipeline well executed. However, they raised substantial concerns regarding novelty and positioning relative to closely related and concurrent work, evaluation rigor and realism, and the strength and scope of empirical validation, such as:

The absence of real sample.
Unfair comparison of experiments.
Lacking experiments on attacking self-supervised learning (SSL) component or the state of the art detector models.

**Reviewer Concerns:**

Reviewer rpfF’s concern regarding the absence of real samples in the evaluation remains unaddressed. While the authors argue that their goal is not to benchmark detectors, evaluating detection robustness in practice fundamentally requires real samples to contextualize false negatives and to support standard metrics (e.g., EER). In addition, the reviewer explicitly requested comparisons against in-the-wild fake datasets such as In-the-Wild and Deepfake-Eval-2024 to assess realism and fairness. These additional experiments were not provided, and the response largely reiterates the original experimental design without resolving the concern about unfair or incomplete comparisons.

Reviewer 6vBc’s concern about the lack of experiments on detectors with self-supervised learning (SSL) components is only partially addressed. Although the authors expanded the evaluation to include additional open-source detectors, the rebuttal does not convincingly demonstrate coverage of modern SSL-based ADD architectures or analyze whether the proposed attack behaves differently in this regime.

Finally, in response to reviewer uoRF, the authors did provide additional detector results in the rebuttal, which help clarify the effectiveness of the method on stronger open-source models. However, these additions do not fully compensate for the broader concerns regarding evaluation completeness and realism raised by the other reviewers.

**Reviewer Scores:**

6vBc and rifF's review will remain negative given the weakness and the extend it has been answered.

---

### Decision · Program_Chairs · 2026-01-26

Reject